# Mechanisms of Action of Fruit and Vegetable Phytochemicals in Colorectal Cancer Prevention

**DOI:** 10.3390/molecules28114322

**Published:** 2023-05-24

**Authors:** Teresita Alzate-Yepes, Lorena Pérez-Palacio, Estefanía Martínez, Marlon Osorio

**Affiliations:** 1School of Nutrition and Dietetics, University of Antioquia, Carrera 75 # 65-87, Medellín 050010, Antioquia, Colombia; teresita.alzate@gmail.com (T.A.-Y.); lorena.perezp@udea.edu.co (L.P.-P.); 2School of Engineering, Pontifical Bolivarian University, Circular 1 No. 70-01, Medellín 050031, Antioquia, Colombia; estefania.martinezc@upb.edu.co; 3Systems Biology Group, School of Health Sciences, Pontifical Bolivarian University, Calle 78 B # 72 A 10, Medellín 050034, Antioquia, Colombia

**Keywords:** phytochemicals, chemoprevention, colorectal diseases, diet

## Abstract

Colorectal cancer (CRC) is the third most common cancer worldwide and its incidence is expected to increase by almost 80% by 2030. CRC apparition is related to poor diet, mainly due to low consumption of phytochemicals present in fruits and vegetables. Hence, this paper reviews the most promising phytochemicals in the literature, presenting scientific evidence regarding potential CRC chemopreventive effects. Moreover, this paper reveals the structure and action of CRC mechanisms that these phytochemicals are involved in. The review reveals that vegetables rich in phytochemicals such as carrots and green leafy vegetables, as well as some fruits such as pineapple, citrus fruits, papaya, mango, and Cape gooseberry, that have antioxidant, anti-inflammatory, and chemopreventive properties can promote a healthy colonic environment. Fruits and vegetables in the daily diet promote antitumor mechanisms by regulating cell signaling and/or proliferation pathways. Hence, daily consumption of these plant products is recommended to reduce the risk of CRC.

## 1. Introduction

Cancer is defined as a set of functional capabilities acquired by human cells as they evolve from normal to neoplastic growth states, and these capabilities are crucial for the formation of malignant tumors [1]. Lung (12.7%), breast (10.9%), colorectal (9.7%), and stomach (7.81%) cancers are the most diagnosed cancers worldwide. Colorectal cancer (CRC) is the third most common cancer worldwide and is expected to cause about 0.69 million deaths by 2019, with an incidence that continues to increase by 1.5% each year in the population under 50 years of age [2]. The Pan American Health Organization has reported that colorectal cancer is the third leading cause of cancer-related deaths in the Americas, with more than 245,000 new cases each year, and it is estimated that the number of cases will increase by 70–80% by 2030 [3].

Factors such as unhealthy lifestyle, lack of physical activity, smoking, excessive alcohol consumption, and a diet with low amounts of fruits and vegetables are considered to have a considerable impact on the expression and progression of the disease [4]. Although a wide range of procedures are available to treat or remove tumors—radiotherapy, surgical dissection [5,6] and chemotherapy—and are effective in controlling the disease at the primary site, they are also associated with significant disadvantages and side effects, as there remains a possibility that tumors that have been surgically removed can still recur and be resistant to radiation treatment [7]. Hence, it becomes imperative to look for new therapeutic targets and drugs, and importantly, look for efficient ways to prevent such tumors.

In recent times, the gut microbiota (GM) has been an area of interest for the prevention and nutrition-based treatment of chronic noncommunicable diseases (NCDs) due to the close relationship that exists between its balance and the onset or outcome of these diseases. Diet is a major regulator of this balance [8]. Regular consumption of an unhealthy diet is a trigger for dysbiosis and the consequent occurrence of some of the NCDs, while consumption of regulatory substances such as prebiotics, probiotics, and phytochemicals contribute to maintaining the balance. Soluble dietary fiber, mainly composed of inulin, oligosaccharides, and sugars, is the major energy substrate for GM bacteria, whose fermentation in the colon generates volatile fatty acids that can activate specific receptors in numerous tissues and organs of the body. Recent studies have shown that phytochemicals, although consumed in smaller amounts than dietary fibers, also have a positive impact on GM [9], contributing to human health. Thus, it is important to consider them and promote their dietary consumption.

The abovementioned points are supported by the fact that some foods have been gaining the attention of researchers, nutritionists, and food manufacturers in recent years, precisely because they contain compounds such as phytochemicals, which have anticancer properties. Phytochemicals are being increasingly used as convenient diet-additive to improve treatment outcomes in diseases with complex pathological behaviors by affecting numerous cellular targets such as genes and proteins [10,11], and by participating in processes such as inflammation, oxidative stress, and apoptosis, mitigating their incidence and progression, together with acting as natural chemopreventive compounds.

Among the foods with chemopreventive properties, are fruits such as apples, blackberries, cherries, passionfruit, pineapples, mangos, grapes, berries, and blueberries; vegetables such as cauliflower, broccoli, bell pepper, and tomato; oilseeds such as soybeans, peanuts, pistachios, and pine nuts; oils such as olive oil; and some spices such as turmeric and pepper. Each of these has one or several chemopreventive compounds, which can act on the cancer cell through various mechanisms, such as modulating the signaling pathways to promote apoptosis or prevent cell proliferation [12,13].

This thematic review explores the nutritional composition and the content of bioactive compounds in some fruits and vegetables, as well as their chemopreventive effects in terms of their antioxidant and anti-inflammatory properties. This information provides the basis for and allows us to understand the reasons for including such foods in the daily diet, as chemopreventive therapeutic molecules against colorectal cancer or as a coadjuvant in its treatment.

## 2. Methods and Data Collection

Google Scholar, Science Direct, Scopus, and Springer databases were utilized to search information for this review on the molecular mechanisms of cancer prevention associated with phytochemicals, found mainly in foods such as fruits and vegetables.

For the search, the following research term was used “phytochemicals” AND “anticancer mechanism” and “cancer”. In Springer, 2845 references were found, then applying the filters type of source (article), and year of publication (from 2019 to 2023), the number of documents were reduced to 1453. After that, the research was fitted to “colorectal cancer”, leaving only 366 research articles. Finally, the research was filtered by “phytochemicals from fruits and vegetables”, which reported 110 articles. A similar process was used with the Scopus database, finding 1031, 506, 271, 29 and 5 articles, respectively, and in Science Direct the results in a number of documents were: 11,509, 8471, 4664, 1259 and 383, respectively, and in Google scholar, including the article filters, it reported 33,300, 17,400, 15,400, 768 documents, respectively.

Subsequently, the results were contrasted to look for concurrences in the reports and the similarity of contents in the databases with respect to phytochemicals until reaching the 98 references of this article.

The search was not limited to specific study designs, and the relevant literature known to the authors was added to complement the writing of this article. This review focuses on the phytochemicals found in a wide range of foods and their mechanisms of anticancer action. We are seeking for scientific arguments to promote the increase in consumption of those foods, thereby helping prevent colorectal cancer through knowledge-based methods.

## 3. Phytochemicals in Fruits and Vegetables and Their Actions against CRC

A chronic inflammatory state in individuals with malnutrition becomes a predisposing factor for NCDs, including cancer [14]. This is caused by changes or increases in free radicals, such as superoxide anion, hydrogen peroxide, and hydroxyl ions, which accelerate cellular oxidative processes. An increase in oxidative stress can cause long-term cell damage by modifying the functions of the immune and endocrine systems, which are the main metabolic regulators [15].

The immune and endocrine systems perform regulatory functions by secreting hormones such as insulin, estrogen, prolactin, and leptin. Conversely, they can increase the production of tumorigenic factors such as IGF1-1, cytokines, and adipokines in ‘dysregulated’ conditions or in diseases such as obesity. This can generate biochemical cascades, leading to cell damage [16], thereby favoring uncontrolled growth and proliferation of malignant cells. Several studies have reported antioxidant properties of certain types of high nutritional value foods, such as fruits and vegetables, that help in maintaining and regulating oxidative stress that can prevent some chronic diseases [17,18,19].

Moreover, genetic mutations are common causes of most cancers, as are some epigenetic factors. Certain mutations allow cells to evade apoptosis and proliferate uncontrollably; hence, inducing apoptosis in cancer cells has been the gold standard for many therapeutic approaches against this disease, which include the utilization of phytochemical compounds with anticancer properties capable of inducing apoptosis in target cells [20].

The use of synthetic, natural, or biological agents to minimize the occurrence of cancer in healthy individuals is defined as chemoprevention. At the molecular level, chemoprevention has been characterized by changes in multiple pathways, which play a critical role in the three basic steps of carcinogenesis, such as initiation, promotion, and progression [21]. The successful use of chemopreventive agents to protect high-risk populations from cancer, or to improve the therapeutic outcomes in patients with the disease already undergoing chemotherapy, demonstrates that the strategy is rational and promising. Dietary components widely known as phytochemicals are plant secondary metabolites that protect plant tissues against stress or external attacks. Some of these are part of important research areas regarding their roles in human health and disease, specifically, those that inhibit inflammation, reduce oxidative stress, and prevent or delay oxidation by scavenging free radicals in addition to other anti-cancer mechanisms, and which thereby behave as chemopreventive agents [22,23,24].

It is important to note that this review is focused on the effects of natural phytochemicals present in the forms and concentrations found in foods and not those in dietary supplements or drugs, as part of a varied and balanced diet based on natural foods. The effects of dietary supplements containing higher concentrations of phytochemicals may be different. Therefore, it is important to consult a health professional before initiating the consumption of dietary supplements or large amounts of foods rich in phytochemicals. Some phytochemicals found in fruits and vegetables and their mechanisms of action concerning their CRC anticancer effects are described below:

### 3.1. Phenolic Acids and Flavonoids

Phenolic acids (PA) are the precursors of several phytochemicals, including flavonoids and polyphenols. Moreover, they include those substances that have several phenol functions linked to aromatic or aliphatic structures in their structures, and only some phenolic compounds of the family of phenolic acids are monophenols. In general terms, phenolic compounds are classified into simple (monophenols, see Figure 1 upper ) or complex phenolic acids (polyphenols, see Figure 1 lower); the most common monophenols are those derivatives of benzoic acid (vanillic, protocatechuic, syringic, gallic and gentisic acid), found mainly in red fruits, onions and black radishes, and the derivatives of cinnamic acid (p-coumaric, ferulic, caffeic, and cinnamic acid) found in coffee, tea, cocoa, apples, pears, berries, and some whole grain cereals [25]. Complex phenolic acids are found mainly in leafy vegetables, onions, apples, berries, cherries, soybeans and citrus fruits, and their structures and mechanisms of action are explained below. The anticancer potential attributed to phenolic acids is mainly due to their antioxidant activity; however, they have also exhibited anticancer effects associated with their ability to inhibit cell proliferation (extracellular signal-regulated kinase (Erk)1/2, D-type cyclins and cyclin-dependent kinases (CDKs)), see Figure 2. Likewise, PA also has impacts on angiogenic factors (vascular endothelial growth factor (VEGF) and MIC-1), oncogenic signaling cascades (phosphoinositide 3-kinase (PI3K) and protein kinase B (Akt)), which induce apoptosis and prevent cell migration and metastasis [26].

Polyphenols and flavonoids (complex phenolic acids) can be easily confused with each other due to their structural similarities and phytochemical properties. However, polyphenols are a large family of compounds (>8000) with more than one phenolic group in their chemical structure [27] and are of natural, semi-synthetic, or synthetic origin [28]. Flavonoids are a group of complex phenolic acids that share a common structure of 15-carbon skeleton and are of natural origin [29]. The basic structures of phenolic acids and flavonoids, and their derivatives that are suggested to have chemopreventive properties are shown in Figure 1. Table 1 reviews the foods with these phytochemicals with chemoprevention activity evaluated in in vitro and in vivo scenarios.

The mechanism of action of polyphenols and flavonoids against cancer cells share a common process that involves two pathways, the first one is via mitogen-activated protein kinases (MAPK), and the second is mediated by anti-inflammatory properties through binding of tumor necrosis factor (TNF) complex to its receptor (TNFR). The first pathway plays an important role in the regulation of physiological processes such as the control of cell growth, development, cell division, and cell death [30,31]. For example, polyphenols, such as quercetin, induce phosphorylation of *p38*, c-Jun N-terminal kinases (JNK), and extracellular signal-regulated kinase (ERK) proteins [32], which disrupt the regulation of Bcl-2, Bcl-xl, and Bax proteins, thereby releasing cytochrome c and inducing cell apoptosis in tumor cells. In the second pathway, polyphenols, such as curcumin and isoflavones, act by inhibiting TNF and TNFR complexes and *p65* and *p50* gene expression, respectively, thereby inducing an anti-inflammatory effect by decreasing interleukin-1β (IL-1β), IL-6, IL-8, and tumor necrosis factor-α (TNF-α). In addition, polyphenols can inhibit phosphorylated JNK protein, which induces phosphorylation of kinase IκBα, which also suppresses the induction of TNF-α [33]. The full mechanism is depicted in Figure 2. The specific mechanisms are described in the subsequent sections.

#### 3.1.1. Quercetin

Quercetin (IUPAC name 2-(3,4-dihydroxyphenyl)-3,5,7-trihydry-4H-chromen-4-one) is widely recognized for its high antioxidant, anti-allergic, anti-inflammatory, and antiviral properties. The antiproliferative and proapoptotic effects of quercetin have recently been investigated in tumor cells in the colon. This is a naturally occurring flavonoid that exists widely in fruits and vegetables such as capers, lovage, dill, coriander, onions, apples, and berries [34].

At the cellular level, the targets and mechanisms by which quercetin may affect digestive system cancers include MAPK, adenosine 5′-monophosphate-activated protein kinase (AMPK)/m target of rapamycin (TOR)/*p38*, toll-like receptor 4 (TLR4) or nuclear factor-κB (NF-κB), JNK/transcription factor Jun (JUN), focal adhesion kinase (FAK), protein kinase B (AKB or AKT), mitogen-activated protein kinase kinase (MEK)1/2 and ERK1/2 signaling pathways, E-cadherin proteins, NF-κB, p65, TLR4, matrix metalloproteinases (MMP), tissue inhibitors of metalloproteinases (TIMP), epithelial–mesenchymal transition (EMT) markers (E-catenin, N-catenin, Wnt/β-catenin, and Snail), and specificity proteins (Sp1, Sp2, Sp3, and Sp4) [5,35]. Quercetin can regulate the expression of E-cadherin, NF-κB, p65, anti-TLR4, cannabinoid receptor CB1-R, and EMT markers (E-catenin, N-catenin, Wnt/β-catenin, and Snail). Current studies on the effects of quercetin in colorectal cancer indicate that this flavonoid can inhibit the cell viability of CT26 and MC38 cancer cells and induce apoptosis via MAPK pathways [36]. Similarly, studies have concluded that 8-C-(E-phenylethenyl) quercetin, a quercetin derivative, triggers G2 phase arrest in colorectal cancer cells and inhibits proliferation. It also induces autophagic cell death upon stimulation with ERK [37].

In colorectal cancer, many previously unreported proteins, which are involved in various metabolic pathways have been predicted to be potential targets for quercetin. These proteins are associated with cell proliferation, actin filament depolymerization, transcriptional dysregulation, stabilization of the tumor suppressor protein p53, the mitochondrial fatty acid beta-oxidation pathway, metabolism of xenobiotics by cytochrome P450, and mediation of male sexual differentiation. The proteins are described in detail below [5]:

(1) FOS-like antigen 1 (FOSL1) promotes cell proliferation. High FOSL1 expression, has been shown to be associated with a lower survival rate, indicating that quercetin can reduce cancer cell proliferation by reducing FOSL1 expression [38].

(2) V-maf avian musculoaponeurotic fibrosarcoma oncogene homologue (*MAF*) is related to transcriptional misregulation, which can accelerate cancer progression. Quercetin was shown to delay transcriptional dysregulation by reducing MAF expression and thus reducing colorectal carcinogenesis.

(3) Cofilin-2 (CFL2) promotes actin filament depolymerization, which can reduce cell stiffness [39]. Quercetin was predicted to reduce CFL2 expression to enhance cell stiffness and thus reduce colorectal carcinogenesis.

(4) Cyclin-dependent kinase inhibitor 2A (CDKN2A) has a product that stabilizes the tumor suppressor protein p53 since it can interact with, and sequester the E3 ubiquitin-protein ligase MDM2, a protein that mediates p53 degradation. The *CDKN2A* gene is frequently mutated in a variety of tumors. Quercetin probably functions at a later stage by promoting CDKN2A expression.

(5) Acyl-CoA dehydrogenase, short chain C-2 to C-3 (ACADS) catalyzes the initial step of the mitochondrial fatty acid beta-oxidation pathway. Quercetin was predicted to prevent fatty acid oxidation by reducing ACADS expression and ultimately reducing colorectal carcinogenesis.

(6) Anti-Müllerian hormone (AMH) is a member of the transforming growth factor-beta gene family, which is involved in male sexual differentiation. Quercetin was predicted to reduce AMH levels and thus reduce the risk of colorectal cancer by downregulating *AMH* gene expression.

#### 3.1.2. Isoflavones

Soy and soy products are the richest sources of isoflavones in the human diet, and epidemiological studies have also shown the beneficial effects of soy isoflavones on breast, prostate, and colorectal cancer in countries such as China and Japan, which have high dietary intake of soy isoflavones [40].

Isoflavones are considered as chemical compounds derived from heterocyclic phenols, that have a structure very similar to the structure of estrogens and include genistein (IUPAC name 5,7-Dihydroxy-3-(4-hydroxyphenyl)chromen-4-one), daidzein (IUPAC name 7-Hydroxy-3-(4-hydroxyphenyl) chromen-4-one), and glycitein (IUPAC name 7-Hydroxy-3-(4-hydroxyphenyl)-6-methoxy-4-chromenone). There is growing evidence showing that isoflavones exert antitumor activities by regulating cell proliferation, apoptosis, cell cycle, migration, invasion, angiogenesis, and metastasis by modulating several signaling pathways, including Akt, NF-κB, and Notch.

Furthermore, genistein is proposed as a promising chemopreventive agent in cancer treatment [41,42,43,44,45]. It reduced the CD133^+^CD44^+^ subpopulation in HCT116 colorectal cancer cells, which was correlated with the upregulation of PTEN and downregulation of FASN expression [44,46]. In addition, genistein in combination with metformin, an antidiabetic drug, and lunasin, a bioactive component of soybean, increased the sensitivity of CD133^+^CD44^+^ cells to 5-fluorouracil (5-FU). Genistein consistently reduced the expression of CD133, CD44, and β-catenin in 1,2-dimethylhydrazine (DMH)-induced colorectal cancer in rats, suggesting that genistein might inhibit DMH-induced colon cancer stem cells (CSCs) [46,47].

#### 3.1.3. Resveratrol

Resveratrol (IUPAC name 5-[(E)-2-(4-hydroxyphenyl) ethenyl]benzene-1,3-diol) is a naturally occurring stilbenoid polyphenol, often derived from peanuts, grapes, pistachios, red wine, blueberries, and verjuice. Resveratrol has been shown to increase the release of several proinflammatory cytokines from immune cells, thereby promoting anti-cancer cytotoxicity [48]. In addition, it exhibits antineoplastic effects by targeting CSCs in a wide range of cancers. For example, treatment of colon CSCs with resveratrol promoted the mesenchymal–endothelial transition and reduced cellular resistance by regulating the autophagy pathway and polypeptide N-acetylgalactosaminyltransferase 11 (GALNT11) expression [49]. Moreover, previous studies have also reported that resveratrol augments the anticancer activities of the chemotherapeutic agent oxaliplatin by synergistically enhancing its tumor suppressor properties [50].

Other studies have shown that resveratrol can induce apoptosis and cell cycle arrest in most tumor cells [51,52]. Additionally, in neurons and endothelial cells undergoing severe oxidative stress, resveratrol can also help prevent radiation damage and drug-induced cytotoxicity by reducing oxidative stress and reorganizing the expression of the protein, survivin. Consequently, resveratrol can be used in chemotherapy to protect normal cells and reduce the side effects of anti-cancer drugs.

The specific molecular mechanisms and signaling pathways implicated in the anticancer effects of resveratrol include: regulation of the caspase and mitochondrial cascade enzyme system; upregulation of cyclin-dependent kinase inhibitors, tumor suppressor genes, and cytokines; downregulation of the expression of survival proteins associated with chemical resistance, including survivin, cFLIP, cIAP, and anti-apoptotic proteins (Bcl-2 and Bcl-XL); activation of protein kinase, and inhibition of MAPK, phosphoinositide 3-kinase (PI3K)/Akt, PKC, EGFR kinase, nuclear factor NF-κB, activator protein 1 (AP-1), HIF-1, and signal transducer and activator of transcription 3 (STAT3) [53].

However, the low bioavailability of resveratrol is a major constraint for extrapolating its impacts in human patients [54], which has led to the development of different methods to increase its bioavailability, including using it in combination with the phytochemical quercetin, the food sources of which have been described above. The phytochemicals present in this type of diet mixture, produced by combining the food sources, can play symbiotic roles with one another.

#### 3.1.4. Curcumin

Curcumin (IUPAC name (1E,6E)-1,7-bis(4-hydroxy-3-methoxyphenyl)-1,6-heptadiene-3,5-dione) is mainly found in turmeric and has antioxidant, anti-inflammatory, antiviral, and antifungal actions, with its anticancer potential being the most described [55]. Curcumin has been reported to modulate growth factors, enzymes, transcription factors, kinases, inflammatory cytokines, and proapoptotic (by upregulation) and antiapoptotic (by downregulation) proteins [56]. Therefore, this polyphenol compound, alone or in combination with other agents, could represent an effective alternative for cancer therapy.

Previous studies [46,57] have shown that curcumin in combination with the Src inhibitor dasatinib eliminated chemoresistant colon (CSCs). In addition, curcumin and/or dasatinib treatments in adenomatous polyposis coli Min+/− mice reduced the expression of CD44, CD133, CD166, and aldehyde dehydrogenases, indicating that curcumin and dasatinib could reverse chemoresistance by targeting colon CSCs.

A novel nanoscale curcumin-loaded micelle has been developed and shown to inhibit colonic CSCs and promote anticancer activity both in vitro and in vivo [58]. The AMPK pathway has also gained interest as an important pathway involved in cancer control. Curcumin has been reported to be an inhibitor of colorectal cancer invasion via AMPK-induced inhibition of NF-κB, urokinase-type plasminogen activator (uPA), and matrix metallopeptidase 9 (MMP-9). Additionally, curcumin was shown to induce apoptosis in human colon cancer cells via a mitochondrial-mediated pathway and trigger the release of cytochrome c, significantly increasing Bax and p53, and reducing Bcl-2 and survivin in LoVo cells [59].

#### 3.1.5. Kaempferol

Kaempferol (IUPAC name 3,5,7-trihydroxy-2-(4-hydroxyphenyl)-4H-1-benzopyran-4-one), which is found in broccoli, cabbage, kale, beans, leek, tomato, strawberries, and grapes, has antiproliferative activity in different types of cancer, including colon cancer; this compound has been shown to act as a cytostatic component in the KNC cell line among other cell lines with diverse functions, profiles and gap junction gene expression profiles [60]. Moreover, in the colon cancer cell lines, RKO and HCT-116, treatment with kaempferol has shown inhibition, decreased viability, apoptosis, autophagy, and changes in coding and non-coding gene expression [61]. In addition, the effect of kaempferol treatment on the phases of the cell cycle was investigated, and fewer cells were observed in the G1 phase and more in the G2 phase.

Additionally, treatment with kaempferol decreased glucose consumption, lactic acid accumulation, and ATP production, thereby indicating an inhibitory effect on tumor aerobic glycolysis, the main source of energy in these tumor cells [62].

Other data suggests that kaempferol may play an important role in overcoming resistance to therapy with 5-FU, a chemotherapeutic drug commonly used in the treatment of colorectal cancer, by regulating the miR-326-hnRNPA1/A2/PTBP1-PKM2 axis [63].

#### 3.1.6. Epigallocatechin-3-Gallate

Tea polyphenols, such as epigallocatechin-3-gallate (EGCG; IUPAC name (2R,3R)-5,7-Dihydroxy-2-(3,4,5-trihydroxyphenyl)-3,4-dihydro-2H-1-benzopyran-3-yl 3,4,5-trihydroxybenzoate), has been shown to modulate cell signaling pathways, regulate cell proliferation, and exhibit antitumor, anti-inflammatory, proapoptotic, and angiogenic activities [64]. EGCG also exhibits antioxidant activity through catalytic metal chelation, and hydrogen atom and electron transfers.

One study showed that EGCG decreased the number of aberrant crypt foci and colorectal tumors in rats [65]. Another study assessed the combined effect of green tea and sodium butyrate on the regulation of survivin, an antiapoptotic protein overexpressed in colorectal cancer cells, and found that treatment with the mixture induced apoptosis and cell cycle arrest at the G2/M and the G1 phase in colorectal cancer cells [66]. It has also been shown to improve sensitivity to 5-FU in spheroid-derived CSCs (SDCSC) [67].

#### 3.1.7. Other Polyphenols

Studies in humans, animals, and cell culture systems suggest that phenols in extra-virgin olive oil have a protective effect against different types of cancer [68]. This characteristic is mainly attributed to the antioxidant properties of the phenolic compounds, particularly hydroxytyrosol (3,4-dihydroxyphenylethanol), tyrosol (p-hydroxyphenylethanol), pinoresinol, caffeic acid, oleuropein, lignans, and squalene [69,70], as well as some vitamins, such as D (cholecalciferol), E (tocopherol), A (retinol), and K (phylloquinone) [71].

In human colorectal cancer cells, extra-virgin olive oil promoted the expression of the cannabinoid receptor type 1 (*CNR1*) gene encoding the cannabinoid receptor 1 (CB1) and suppressed cell proliferation [72]. It should be noted that CB1 is expressed in different tissues under pathological conditions, including colorectal cancer.

Both hydroxytyrosol and oleuropein induce cell cycle arrest and death in colorectal cancer cells, and, therefore, extra-virgin olive oil can be considered effective against colorectal cancer [73]. However, prospective cohort studies or large-scale case-control studies are required to assess the association between the consumption of virgin olive oil and its effect on colorectal cancer risk [74], even in regions following a Mediterranean diet, which use olive oil extensively.

Whole grain sorghum has shown important health benefits, having antimicrobial properties associated with its participation in the elimination of free radicals and reduction in oxidative stress, and by its anticancer and anti-inflammatory activities [75] due to presence of bioactive nutrients such as vitamin B, fat-soluble vitamins (D, E, and K), micro and macronutrients, as well as non-nutrient components such as polyphenols (phenolic acids, flavonoids, and condensed tannins) [76,77]. Sorghum has been shown to contain several compounds with potential anticancer properties. These 3-Deoxyanthocyanins in black sorghum have been shown to possess both anticancer and antioxidant properties in vitro. Apart from 3-deoxyanthocyanins, sorghum also contains flavones with estrogenic properties, which have an anticancer effect in vitro [78]. The mechanisms underlying the inhibition of cancer cell growth by sorghum phenolic compounds are now clear, and demonstrate their cytostatic capacity and proapoptotic activity in various cancer cell lines [79]. Table 1 reviews the foods with phytochemicals with chemoprevention activity evaluated in in vitro and in vivo scenarios.

**Table 1 molecules-28-04322-t001:** Phytochemicals presented in food.

Phytochemical	Food	Type of Effectiveness Evaluation	References
Phenolic acids	Berries, onions, black radishes, coffee, tea, cocoa, apples, pears, berries, and whole grain cereals	In vivo/In vitro	[25,26,76,77,79,80,81]
Resveratrol	Peanuts, grapes, pistachios, red wine, blueberries, and verjuice	In vivo/In vitro	[12,48,49,52,54,82]
Flavonoids	Leafy vegetables, onions, apples, berries, cherries, soybeans, and citrus	In vivo/In vitro	[12,31]
Curcumin	Turmeric	In vivo/In vitro	[12,46,55,56,57,58,59]
Quercetin	Capers, lovage, dill, coriander, onions, apples, and berries	In vivo/In vitro	[5,32,34,35,36,37,82]
Kaempferol	Broccoli, cabbage, kale, beans, leek, tomato, strawberries, and grapes	In vivo/In vitro	[36,60,61,62,63]
Epigallocatechin-3-gallate	Green tea	In vitro/In vitro	[31,64,65,66,67,82]
Isoflavones	Soy and soy products	In vitro/In vitro	[33,40,41,43,44,46,47]
Hydroxytyrosol, tyrosol, pinoresinol, oleuropein, lignans, and squalene	Extra-virgin olive oil	In vitro/In vitro	[68,69,70,72,73,74]
Vitamins B, D, E and K	Sorghum, extra-virgin olive oil	In vitro/In vitro	[75,76,83]
Carotenoids	Sorghum, tomato, carrot, pineapple, citrus fruits, papaya, sunflower flower, saffron, and green leaves, and other fruits and vegetables with pigments yellow, orange, and red	In vitro/In vitro	[75,76,77,80,81,82,84,85,86]
Piperine	Black, green, and white pepper	In vitro/In vitro	[84,87,88,89,90]
Sulforaphane	Cruciferous vegetables, including cauliflower and broccoli	In vitro/In vitro	[82,84,91,92]

### 3.2. Other Chemopreventive Compounds

Apart from polyphenolic compounds, other phytochemicals, such as alkaloids (including piperine), sulforaphanes, and carotenoids, also occur in nature. Their CRC chemopreventive mechanism is shown in Figure 2, their chemical structures are shown in Figure 3, and Table 1 reviews the foods with phytochemicals with chemoprevention activity evaluated in in vitro and in vivo scenarios. Sulforaphane and piperine induce the production of free radicals (such as reactive oxygen species (ROS)) in the extracellular space. High levels of ROS trigger the activation of MAPK pathways, with activation of ERK and JNK proteins, which in turn induce p21 and downregulate cyclin D1, causing G1-phase cell cycle arrest, and, therefore, induce apoptosis [87]. The exact mechanism of action of carotenoids is not fully understood and should be explored in future studies.

#### 3.2.1. Piperine

Piperine (IUPAC name 1-[5-[1,3-benzodioxol-5-yl]-1-oxo-2,4-pentadienyl]pi-peridine) is a nitrogen-containing alkaloid molecule found naturally in black, green, and white pepper of the Piperaceae family, with its content ranging from 2 to 7.4% in black and white pepper. Piperine has gained considerable interest over the past two decades for its beneficial health effects [93]. Clinical trials have investigated the protective and therapeutic effect of piperine against many diseases and disorders, including cancer, and its pleiotropic mechanistic action is attributed to its ability to interact with a broad spectrum of molecular targets, including kinases, transcription factors, cell cycle proteins, inflammatory cytokines, receptors, and signaling molecules [94].

A preclinical study has suggested that piperine acts on several cell cycle proteins (cyclin D, cyclin T, CDK2, and CDK4) and exerts an anticancer effect by inhibiting S phase and forming a hydrogen bond with Ser5 at the ATP binding site of CDK2 [95]. MMP-9 is abundantly expressed in malignant tumors and contributes to cancer invasion and metastasis. PKCα/ERK1/2 and NF-κB/AP-1 pathways are major signaling pathways that regulate cancer cell invasion. In the study, piperine was shown to downregulate MMP-9 expression by inhibiting PKCα/ERK1/2 and NF-κB/AP-1 pathways in a PMA (phorbol-12-myristate-13-acetate)-induced cancer model in vitro. It also inhibited HT-1080 cell invasion and migration. Moreover, GM-CSF, TNF-α, MMP-2, MMP-9, and proinflammatory cytokines such as IL-1β and IL-6 are involved in NF-κB and AP-1-mediated cancer progression. Piperine inhibited the translocation of NF-κB subunits such as p50, p65, and c-Rel, as well as CREB, ATF-2, and c-Fos [96].

Wnt/β-catenin signaling is a molecular target for colorectal, ovarian, and hepatocellular cancer. Piperine inhibits Wnt/β-catenin signaling by affecting the binding of T-cell factor (TCF) to DNA, thus affecting cell cycle progression [88,97]. Piperine in combination with curcumin has been reported to cause a six-fold increase in caspase-3 activity and to be more efficient in inhibiting cell proliferation [90]. It also decreases metastasis of intestinal tumor cells [89]. In addition, piperine was found to inhibit cell cycle progression in rectal cancer cells by inducing ROS-mediated apoptosis [98].

#### 3.2.2. Carotenoids

Carotenoids are a group of pigments found mainly in tomato, carrot, pineapple, citrus fruits, papaya, sunflower flower, saffron, and green leaves, and provide the yellow, orange, and red colors in many plants. Vegetable leaves are a vital source of carotenoids and mainly include α (IUPAC name 1,3,3-trimethyl-2-{(*1E*,3*E*,5*E*,7*E*,9*E*,11*E*,13*E*,15*E*,17*E*)-3,7,12,16-tetramethyl-18-[(1*R*)-2,6,6-trimethylcyclohex-2-en-1-yl]octadeca-1,3,5,7,9,11,13, 15,17-nonaen-1-yl}cyclohex-1-ene) and β (IUPAC name 1,3,3-trimethyl-2-[3,7,12,16-tetramethyl-18-(2,6,6-trimethylcyclohex-1-en-1-yl)octadeca-1,3,5,7,9,11,13,15,17-nonaen-1-yl]cyclohex-1-ene) carotene, as well as other carotenoids, such as lutein, neoxanthin, crocetin, antheraxanthin, and violaxanthin, which are found in very low amounts. Experimental studies have identified several mechanisms through which carotenoids control the development of various cancers in humans [84]. These mechanisms include cell signaling and cell communication functions, and antioxidant activities, which reduce cancer risks [85]. For example, β-carotene, β-cryptoxanthin, lycopene, lutein, and zeaxanthin can be used as chemopreventive agents. β-cryptoxanthin also regulates the expression of the *RB* gene, which is a known anti-oncogene. Moreover, carotenoids extracted from vegetables have protective, preventive, and even curative effects against various types of cancer [99].

With cancer being associated with inflammatory processes, the beneficial effects of both lutein and zeaxanthin are due to their anti-inflammatory and antioxidant properties, though the exact mechanism of action of both compounds is not fully understood and needs to be explored further in the future. Carotenoids can use various pathways for their anticancer activity, but their antioxidant function is considered the most common. A recent review [86] explored the relationship of the carotenoids obtained through diet such as lycopene, α- and β-carotene, lutein, zeaxanthin, and β-cryptoxanthin, with some of the most common cancers, and found colorectal cancer to be among them [82].

Yellow passionfruit (*Passiflora edulis flavicarpa*), generally consumed as juice, contains phytochemicals (mainly carotenoids) that have potential benefits for human health. A study has found that passionfruit juice extract activates caspase-3 in HL-60 and MOLT-4 cells in a BALB/c 3T3 neoplastic transformation model. This indicates that the chemical components of the passionfruit juice extract can induce apoptosis [80], and these beneficial results were obtained with levels that could theoretically occur in the plasma after consumption of the juice.

The variety of carotenes in other lesser known and less consumed tropical fruits, such as lychee (*Lychee chinensis*), açaí (*Euterpe oleraceae*), and mamey sapote (*Pouteria sapota*), have also been reported as pertinent phytochemicals with antioxidant and anti-inflammatory activities, and thus may feature as highly relevant compounds that can minimize the risk of cancer [90].

#### 3.2.3. Sulforaphane

Finally, sulforaphane (SFN; IUPAC name 1-isothiocyanato-4-(methanesulfinyl) butane) is a dietary component obtained from cruciferous vegetables, including cauliflower and broccoli. It acts as a chemoprotective agent and targets CSCs in a variety of human malignancies by promoting programmed cell death/apoptosis, arresting the cell cycle, inhibiting angiogenesis, reducing inflammation, altering susceptibility to carcinogens, and reducing invasion and metastasis [91]. SFN can arrest cancer cells in the G1 or G2/M phases in different cell lines, mainly intervening in the G2/M phase (Figure 2). For example, in HCT-116, HT29, and Caco-2 human colon cancer cells, this type of cell cycle arrest results in senescence. It should be noted that the anticancer and antioxidant potential of SFN is mainly due to its high content of glucosinolates (GSL) [92].

## 4. Conclusions and Future Aspects

Health problems are increasing worldwide, mostly due to chronic rather than acute diseases, with NCDs, like cancer, accounting for more than half of the global disease burden. Hence, prevention strategies with a global approach are a priority, based on the knowledge about risk factors that, like diet, are an integral part of the lifestyle [100]. Knowing the components of food and the compounds that can prevent, reduce, or control cell damage is important to promote food intake behaviors, which can lead to the consumption of these compounds.

A diet rich in fruits and vegetables (recognized as the main sources of phytochemicals) can contribute to a decrease in cancer rate, as indicated by different studies. Thus, it is important to increase the consumption of fruits and vegetables in different meals throughout the day, since they decrease the risk of inflammation and oxidative stress, and can therefore help prevent different types of cancer, including colorectal cancer, in addition to gastrointestinal discomfort such as constipation (a precursor of multiple diseases). Due to the presence of bioactive compounds, fruits, and vegetables boost the immune system, thereby reducing the risk of multiple diseases and their complications. Therefore, consumption of foods that provide different bioactive compounds is recommended: coriander, onions, and apples for their quercetin content; grapes, verjuice, and mortino berries for resveratrol; cruciferous vegetables, including cauliflower, broccoli, cabbage, and radish for sulforaphane; a wide range of vegetables such as tomato, carrot, and several leafy greens, as well as various fruits, including pineapple, citrus fruits, papaya, mango, cape gooseberry, *mortiño* berries, verjuice, passionfruit, and strawberry for carotenoids; turmeric for its contribution of curcumin, and white pepper for its piperine. Accordingly, it is recommended, if possible, to prefer the consumption of olive oil due to its high content of phenolic compounds. The consistency of results for colorectal cancer risk across different populations, suggests that consuming a dietary pattern that is high in fruits and vegetables and low in meats and sweets is protective against colorectal cancer development.

Finally, based on the findings referred to here, future research in the subject should be carried out using relevant scientific evidence to demonstrate the benefits of fruit and vegetable phytochemicals that could have anticancer effects in the colon. The food reviewed here (with contents of phytochemicals) are healthy alternatives for consumption in contrast to ultraprocessed products that contribute directly to the appearance of cancer in colon, and also broaden the spectrum for colorectal cancer chemoprevention.

## Figures and Tables

**Figure 1 molecules-28-04322-f001:**
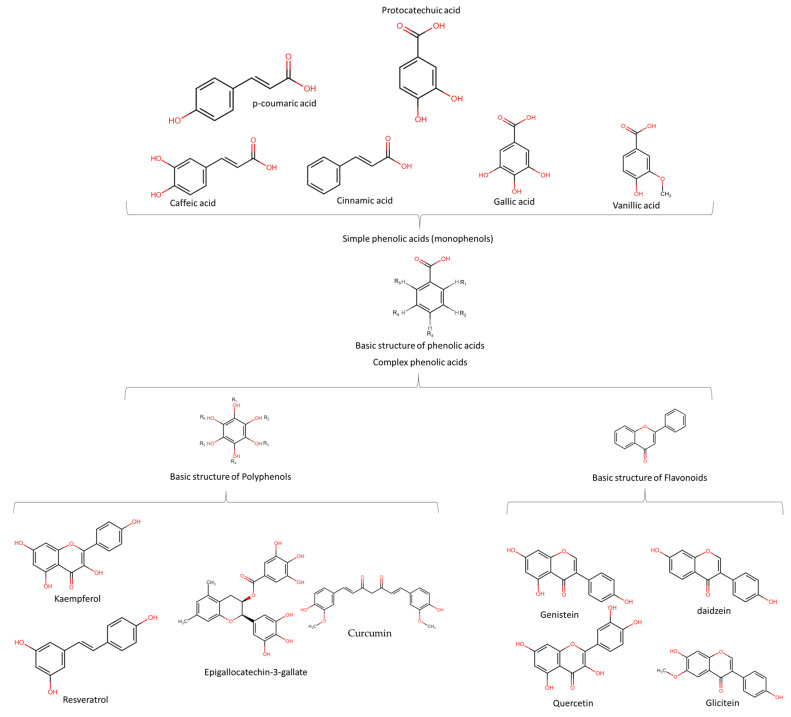
Basic structures of phenolic acids, polyphenols and flavonoids, and their derivatives implicated as chemopreventive compounds.

**Figure 2 molecules-28-04322-f002:**
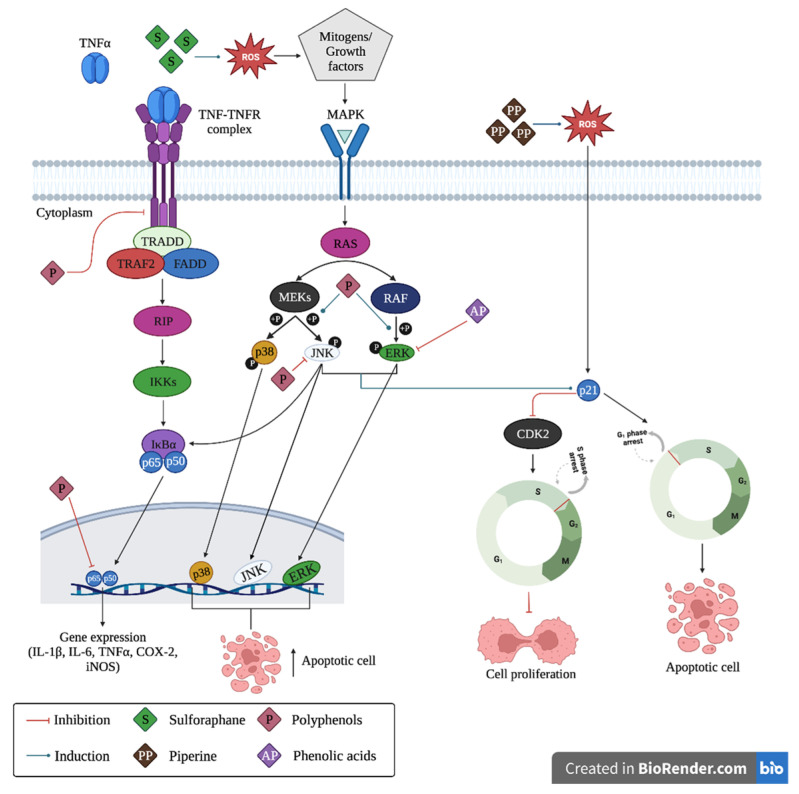
Mechanisms of anticancer action of phytochemicals for colorectal cancer chemoprevention. TNFα: tumor necrosis factor, TNF-TNFR: tumor necrosis factor-tumor necrosis factor receptor complex, TRADD: tumor necrosis factor receptor type 1, TRAF2: TNF receptor-associated factor 2, FADD: FAS-associated death domain protein, RIP: ribosome-inactivating protein, IkBa: inhibitor of nuclear factor kappa-B kinase subunit alpha, p65 and p50: Transcription factors 65 and 50, respectively, ROS: reactive oxygen species MAPK: mitogen-activated protein kinase, RAS: guanosine-nucleotide-binding protein, MEKs: mitogen-activated protein kinase kinase, RAF: serine/threonine-specific protein kinases, p38: mitogen-activated protein kinases, JNK: c-Jun N-terminal kinase, ERK: extracellular signal-regulated kinase, P21: cyclin-dependent kinase inhibitor, CDK2: cell division protein kinase 2.

**Figure 3 molecules-28-04322-f003:**
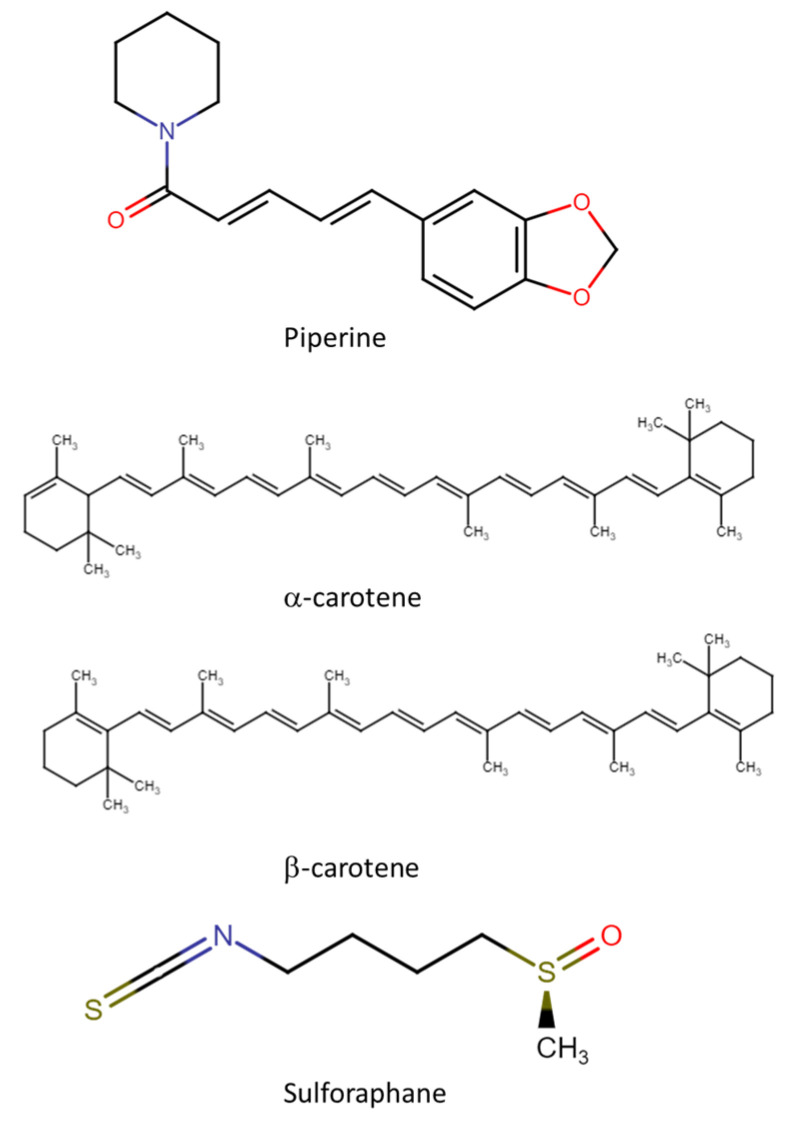
Chemical structures of non-polyphenolic compounds.

## Data Availability

Not applicable.

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
