# Peer review of "Mechanisms of Action of Fruit and Vegetable Phytochemicals in Colorectal Cancer Prevention"

_molecules, 2023, doi:10.3390/molecules28114322_

Round 1

Reviewer 1 Report

The article is very topical and well-written. Only flavonoids are mentioned in the section on phenolic compounds. Phenolic acids, which are a subgroup of phenolic compounds, can be added to those that are important to be taken as food.

Author Response

On behalf of the authors of the paper entitle: Mechanisms of action of fruit and vegetable phytochemicals in colorectal cancer prevention”. Summited to the International Journal of Molecules

Reviewer 1

Comment

Respond

The article is very topical and well-written. Only flavonoids are mentioned in the section on phenolic compounds. Phenolic acids, which are a subgroup of phenolic compounds, can be added to those that are important to be taken as food.

Thank you for the suggestion, phenolic acids were added to the document, along with its action mechanism, structures and food with these phytochemicals (please see New Figure 1, Figure 2 and 3, and table 1)

All changes were highlighted in red in the main document. We hope all comments were properly assessed. Please do not hesitate to contact me if further changes or comments are needed.

Sincerely,

Marlon Andrés Osorio Delgado

Professor of Nanotechnology Engineering

School of Engineering and Health Science

Universidad Pontificia Bolivariana

Circular 1, No. 70-01, Bloque 11

Medellín (Colombia)

Reviewer 2 Report

The subject treated in the article is current and of great interest in the context in which the consumption of vegetables and fruits is quite low and significantly exceeded by highly processed foods lacking valuable nutrients.

To increase the quality of the article, a series of modifications is needed:

First of all, the method of data collection is not fully described with the presentation of the number of studied articles as well as their distribution among the categories of scientific sites investigated.

Secondly, there is a need for a systematization of scientific data in tables that highlight the bioactive compounds from fruits and vegetable products involved in the prevention of colorectal cancer, the type of clinical studies carried out, the dose administered, the changes produced in the human body or in laboratory animals ( highlighting important markers and biochemical parameters). A systematization of the scientific information supported by clinical studies in the form of tables is particularly useful for the argumentation and easier understanding of the impact of bioactive compounds from vegetables and fruits on the prevention of colorectal cancer.

Author Response

On behalf of the authors of the paper entitled: Mechanisms of action of fruit and vegetable phytochemicals in colorectal cancer prevention”. Summited to the International Journal of Molecules I kindly respond to reviewer comments.

Reviewer 2 The subject treated in the article is current and of great interest in the context in which the consumption of vegetables and fruits is quite low and significantly exceeded by highly processed foods lacking valuable nutrients. To increase the quality of the article, a series of modifications is needed:

First of all, the method of data collection is not fully described with the presentation of the number of studied articles as well as their distribution among the categories of scientific sites investigated.

The information of the data collection was added to the section 2. Methods and Data Collection

Secondly, there is a need for a systematization of scientific data in tables that highlight the bioactive compounds from fruits and vegetable products involved in the prevention of colorectal cancer, the type of clinical studies carried out, the dose administered, the changes produced in the human body or in laboratory animals (highlighting important markers and biochemical parameters). A systematization of the scientific information supported by clinical studies in the form of tables is particularly useful for the argumentation and easier understanding of the impact of bioactive compounds from vegetables and fruits on the prevention of colorectal cancer.

A new table (Table 1) was added to the document. It reviews the foods with the discussed phytochemicals across the paper. Moreover, the table demonstrates the scenarios in vitro (cell evaluations) and in vivo (animal/human studies) where the phytochemicals has been studied.

All changes were highlighted in red in the main document. We hope all comments were properly assessed. Please do not hesitate to contact me if further changes or comments are needed.

Sincerely,

Marlon Andrés Osorio Delgado

Professor of Nanotechnology Engineering

School of Engineering and Health Science

Universidad Pontificia Bolivariana

Circular 1, No. 70-01, Bloque 11

Medellín (Colombia)

Reviewer 3 Report

Dear Editors and authors, 

1- The abstract needs to be rewritten again and some information added to show and explain the manuscript more clearly.

2-Figures 2 and 3 contain many symbols that must be explained and interpreted below the figure, in order for the figure to be more clear to the reader.

3-A paragraph should be added on future studies of the use of plant foods in the treatment of cancers of the gastrointestinal tract and colon.

Dear Editors, 

The language of the manuscript is good and acceptable.

Author Response

On behalf of the authors of the paper entitled: Mechanisms of action of fruit and vegetable phytochemicals in colorectal cancer prevention”. Summited to the International Journal of Molecules I kindly respond to reviewer comments.

Reviewer 3

The abstract needs to be rewritten again and some information added to show and explain the manuscript more clearly.

The abstract section was rewritten. Thank you for the suggestion.

Figures 2 and 3 contain many symbols that must be explained and interpreted below the figure, in order for the figure to be more clear to the reader.

The symbols of figures 2 and 3, are clearly defined in the figure legend.

A paragraph should be added on future studies of the use of plant foods in the treatment of cancers of the gastrointestinal tract and colon

The paragraph was added at the end of the conclusions.

All changes were highlighted in red in the main document. We hope all comments were properly assessed. Please do not hesitate to contact me if further changes or comments are needed.

Sincerely,

Marlon Andrés Osorio Delgado

Professor of Nanotechnology Engineering

School of Engineering and Health Science

Universidad Pontificia Bolivariana

Circular 1, No. 70-01, Bloque 11

Medellín (Colombia)

Reviewer 4 Report

Reviewer’s comments

The manuscript (ID: molecules-2382925) reviews the positive effect of phytochemicals, bioactive compounds from fruits and vegetables, on the prevention of colorectal cancer. These phytochemicals show beneficial biological effects on the human body due to their antioxidant, anti-inflammatory, and chemopreventive properties that can regulate the intestinal microbiota. The daily diet rich with phytochemicals contained in fruits and vegetables promotes antitumor mechanisms by regulating cell signaling and proliferation pathways with a focus on reducing the risk of diseases like colorectal cancer and its prevention. The study is well-designed to meet the goals. It is obviously challenged by the types of data accumulated.

The authors are recommended to address the following comments during the revision:

1.     The manuscript should be Grammarly corrected. The recommended corrections have been noticed in the pdf file of the manuscript.

2.     Does the consumption of phytochemicals through fruits and vegetables have the same effect as their direct intake, such as in the form of supplementation, for example?

3.     The study considers only the positive role of phytochemicals from fruits and vegetables in colorectal cancer prevention. However, are there some negative characteristics of phytochemicals for healthy cells or for the human body, in general? Is there any threshold concentration that is toxic?

4.     Although it is a review, I suggest reducing the number of references.

5.     According to the reference list, the authors have not participated in any research related to colorectal cancer and its prevention or the phytochemicals’ effect on some diseases, so the question arises as to why they decided to write a review on the subject.

The manuscript should be Grammarly corrected. The recommended corrections have been noticed in the pdf file of the manuscript.

Author Response

On behalf of the authors of the paper entitled: Mechanisms of action of fruit and vegetable phytochemicals in colorectal cancer prevention”. Summited to the International Journal of Molecules I kindly respond to reviewer comments.

Reviewer 4

The manuscript (ID: molecules-2382925) reviews the positive effect of phytochemicals, bioactive compounds from fruits and vegetables, on the prevention of colorectal cancer. These phytochemicals show beneficial biological effects on the human body due to their antioxidant, anti-inflammatory, and chemopreventive properties that can regulate the intestinal microbiota. The daily diet rich with phytochemicals contained in fruits and vegetables promotes antitumor mechanisms by regulating cell signaling and proliferation pathways with a focus on reducing the risk of diseases like colorectal cancer and its prevention. The study is well-designed to meet the goals. It is obviously challenged by the types of data accumulated.

The authors are recommended to address the following comments during the revision:

The manuscript should be Grammarly corrected. The recommended corrections have been noticed in the pdf file of the manuscript.

The grammar mistakes remarked in the document was corrected. The paper was translated using the service of Elsevier WebShop. Order reference: ASTS0414792.

Does the consumption of phytochemicals through fruits and vegetables have the same effect as their direct intake, such as in the form of supplementation, for example?

Generally speaking, the phytochemicals present in fruits and vegetables turn out to be beneficial for health and their regular consumption has been associated with the prevention of chronic diseases such as colorectal cancer. However, it is important to note that phytochemicals are active chemical compounds that can have both positive and negative effects on the body.

Some phytochemicals can be toxic in large amounts, so it is recommended to consume them in moderate amounts, such as is present in food rather than in supplementation. In addition, some studies have suggested that certain phytochemicals may interfere with nutrient absorption, which could have negative long-term health effects. For example, excessive intake of carotenoids can cause the appearance of a condition called carotenemia, which is characterized by a yellowing of the skin, due to excess carotenoid pigments. However, this is associated with megadoses from the consumption of supplements or drugs, but not from natural foods, since the average in almost all Western countries is lower than that determined by nutritional requirements for all ages.

Regarding the toxicity of phytochemicals, it may depend on the dose and duration of exposure and the interaction with other compounds in the diet. Generally, the phytochemicals present in fruits and vegetables are considered safe in amounts normally consumed as part of a balanced diet. However, some research has suggested that taking high doses of some phytochemicals may have negative health effects. For example, high doses of catechins in green tea can interfere with iron absorption and cause anemia in people with iron deficiency.

It is important to note that most human studies have focused on the consumption of whole foods and that the effects of dietary supplements containing higher concentrations of phytochemicals may be different. Therefore, it is important to consult a health professional before initiating the consumption of dietary supplements or large amounts of foods rich in phytochemicals.

In summary, although the phytochemicals present in fruits and vegetables are generally beneficial for health, it is important to consume them in moderation and as part of a varied and balanced diet based on natural foods.

A proper clarification was added to document lines 141-147,

The study considers only the positive role of phytochemicals from fruits and vegetables in colorectal cancer prevention. However, are there some negative characteristics of phytochemicals for healthy cells or for the human body, in general? Is there any threshold concentration that is toxic?

Although it is a review, I suggest reducing the number of references.

All the references used in the text were considered essential to depict the mechanism and evidence of chemopreventive effect of the phytochemicals treated in the paper. The selection was done according to section 2. Methods and Data Collection

According to the reference list, the authors have not participated in any research related to colorectal cancer and its prevention or the phytochemicals’ effect on some diseases, so the question arises as to why they decided to write a review on the subject.

Papers of the authors were added to the documents. For instance,

M. Londoño-Berrio, C. Castro, A. Cañas, I. Ortiz, and M. Osorio, “Advances in Tumor Organoids for the Evaluation of Drugs: A Bibliographic Review,” Pharmaceutics, vol. 14, no. 12. MDPI, Dec. 01, 2022. doi: 10.3390/pharmaceutics14122709.

P. A. Giraldo Sánchez, K. Jiménez, and T. Alzate Yepes, “Implementation of an Educational Food Intervention in Schoolchildren, Before and During COVID-19 Confinement.,” Perspectivas en Nutrición Humana, vol. 24, no. 1, pp. 85–99, Jul. 2022, doi: 10.17533/udea.penh.v24n1a06.

J. P. Rendón et al., “Evaluation of the Effects of Genistein In Vitro as a Chemopreventive Agent for Colorectal Cancer—Strategy to Improve Its Efficiency When Administered Orally,” Molecules, vol. 27, no. 20, Oct. 2022, doi: 10.3390/molecules27207042.

M. Castaño, E. Martínez, M. Osorio, and C. Castro, “Development of Genistein Drug Delivery Systems Based on Bacterial Nanocellulose for Potential Colorectal Cancer Chemoprevention: Effect of Nanocellulose Surface Modification on Genistein Adsorption,” Molecules, vol. 27, no. 21, 2022, doi: 10.3390/molecules27217201.

All changes were highlighted in red in the main document. We hope all comments were properly assessed. Please do not hesitate to contact me if further changes or comments are needed.

Sincerely,

Marlon Andrés Osorio Delgado

Professor of Nanotechnology Engineering

School of Engineering and Health Science

Universidad Pontificia Bolivariana

Circular 1, No. 70-01, Bloque 11

Medellín (Colombia)

Round 2

Reviewer 2 Report

Accept the revised manuscript in the final form.